# Sport Performance and Manual Therapies: A Review on the Effects on Mitochondrial, Sarcoplasmatic and Ca^2+^ Flux Response

**DOI:** 10.3390/healthcare9020181

**Published:** 2021-02-09

**Authors:** Alex Regno, Attilio Parisi, Marco Chiera, Nicola Barsotti, Claudia Cerulli, Elisa Grazioli, Alessandra Tamburri, Marco Bruscolotti

**Affiliations:** 1Research Commission on Mind-Body Therapies, Italian Society of Psycho Neuro Endocrino Immunology, 00198 Rome, Italy; alex.regno91@gmail.com (A.R.); marco.chiera.90@gmail.com (M.C.); nicola.barsotti@cmosteopatica.it (N.B.); aletamburri@hotmail.it (A.T.); marcobruscolo@tiscali.it (M.B.); 2Department of Movement, Human and Health Science, University of Rome “Foro Italico”, 00135 Rome, Italy; claudia.cerulli@uniroma4.it (C.C.); elisa.grazioli@uniroma4.it (E.G.); 3Italian Triathlon Federation, 00135 Rome, Italy

**Keywords:** mechanotransduction, mitochondria, athletic performance, stress response, manual therapy, calcium ions, bioenergetic metabolism

## Abstract

The present narrative review aims to highlight the possible effects manual therapies could have on cells and mitochondria, as these effects could improve athletic performance management. To this aim, this review summarizes the relationship between mechanical stimulation, with a special focus on physical activity, and cell response based on the most recent mechanobiology findings. Mechanobiology analyzes how cells respond to mechanical stressors coming from the environment. Indeed, endogenous (e.g., blood pressure, heartbeat and gastrointestinal motility) and exogenous (e.g., physical activity and manual therapies) stimuli can induce biochemical and epigenetic modifications that alter protein synthesis with heavy consequences on cell behavior. Mechanical stress can also influence mitochondrial behavior (i.e., biogenesis, autophagy, fusion, fission and energy production), sarcoplasmic response and calcium ion (Ca^2+^) flux. Since manual therapies have been shown to affect the extracellular matrix, which represents a primary source of mechanical stress that may alter both the cytoskeleton and mitochondrial metabolism, it is conceivable manual therapies could also affect cellular and mitochondrial behavior. Lastly, by suggesting possible directions for future laboratory and clinical studies, the authors expect this review to inspire further research on how manual therapies could affect bioenergetic metabolism and, thus, athletic performance.

## 1. Introduction

Manual therapies are a form of treatment delivered by healthcare professionals through their hands to evaluate and treat functional disorders or pathologies [1]. Specific manual techniques are used to engage every body structure, including articulations, myofascial structures, nerve pathways, and circulation. Manual therapeutic approaches such as osteopathic manipulative treatment have shown to be able to improve clinical conditions in both adults and children, sometimes even in a way comparable to standard care [1,2].

In athletics and sports, manual therapies are widely used to prevent injuries, improve symptoms such as post-traumatic muscular pain and myofascial stiffness, and support musculoskeletal rehabilitation [3,4,5,6]. Nonetheless, the biochemical and mechanical effects manual therapies may have on cells remain unclear.

Everyday cells may undergo many mechanical stimuli, either endogenous or exogenous. Examples of endogenous stimuli are the pressures induced by the cardiac sphygmic wave on the blood vessel endothelial cells, or the stimuli alveolar endothelial cells are exposed to during tissue expansion due to breathing mechanics. Exogenous stimuli are, for instance, the manual contact between the therapist and the treated tissue, or the immobilization in case of injuries [7,8,9,10].

Every time cells undergo mechanical stimulations, a complex series of intra- and extra-cellular reactions occurs, which may induce epigenetic effects. In particular, these reactions may cause the cell to express classes of specific genes to regulate the functional response to the mechanical stimulations [7,8,9,10]. In the last few decades, these phenomena have received great attention and have brought about the birth of “mechanobiology.” The process of converting mechanical stimuli into biochemical responses was then defined as “mechanotransduction” [11].

Based on the available literature regarding mechanobiology, sports physiology and manual therapies, the present narrative review focuses on the effects manual therapies could induce at the cellular and molecular levels. In particular, this review aims to assess whether the mechanical stimuli induced by manual therapies could change mitochondrial and sarcoplasmic functions or the calcium ion (Ca^2+^) flux, thus positively influencing bioenergetic metabolism and athletic performance.

Therefore, the present review will discuss the importance of the mechanobiological dimension for muscles, the bioenergetic role of mitochondria, and the mechanosensitive properties of these organelles. Lastly, the main cellular effects shown by manual therapies in preclinical and clinical studies will be described.

## 2. A Brief Summary on Mechanobiology and the Role of Mechanotransduction in Muscles Physiology

Mechanobiology focuses on studying the ways cells respond to the mechanical stimuli they receive from the environment or create during their activity [11]. Whereas complex organisms, such as the human body, are equipped with a connective (fascial), muscular and osteoarticular system, cells have a cytoskeleton that surrounds them and connects each cellular structure from the membrane to the nucleus [12]. In addition, as complex organisms use their musculoskeletal system to move and respond to mechanical stimuli, cells do the same [12]. In fact, mechanical movements do not represent just the macroscopic locomotion, but also the internal and visceral activities such as the heart pulse, the breathing rhythm, the gallbladder contraction, or the movement of immune cells throughout the tissues [12,13,14].

Each cell can respond to mechanical stimuli, and this ability is paramount for every living organism, from microbes to humans, to develop and survive [14,15]. Even plants can show complex responses in the face of mechanical stimuli: for instance, mechanical events may induce plants to experience growth retardation by modifying their hormonal secretion, altering the production of reactive oxygen species (ROS), or changing the activation of the intracellular Ca^2+^-dependent signaling cascade [16,17,18]. In fact, the response to mechanical stimuli could protect plants against biotic and abiotic stressors [16,17,18,19].

The presence of a complex locomotor apparatus made of connective, bone, and muscular tissue gives mechanobiology a unique role in the human body’s physiopathology. A clear example of the importance of mechanobiology is Wolff’s Law, despite its limitations that have been overcome in the years since. Developed in 1892 by the German physician Julius Wolff, the law states that bones adapt to mechanical loads by strengthening their internal and external architecture as mechanical loads increase. On the contrary, as mechanical loads decrease, bone resorption shall occur thus leading to osteopenia or osteoporosis [20]. According to Wolff, bone remodeling depended essentially on how mechanical loads were applied, that is, on their duration, intensity and velocity. Today, we know that bone remodeling occurs since mechanical stimuli are translated into cellular biochemical cascades through the involvement of mechanotransducing processes [21,22].

Another example is the complex remodeling that affects muscles either during training or in case of physical inactivity and immobilization [8,23]. This remodeling involves activating and inhibiting several cellular signaling pathways central to muscle metabolism, such as the mammalian target of rapamycin complex 1 (mTORC1) pathway, the Hippo pathway with its effectors Yap and Taz, and the Wnt pathway [8,23,24,25].

Beyond monitoring the cell’s energetic metabolism, oxidative status and resource availability, the ability to perceive mechanical stimuli and respond to them is paramount for the maintenance and growth of muscular tissue [24,26]. Indeed, biological processes such as protein synthesis, metabolism and muscle growth depend strictly on the mechanical stimuli the muscular system undergoes [24,26,27,28].

In particular, muscle cells change their shape according to the received mechanical stimulus (e.g., an external mechanical force); as a result, muscles can contract, stretch out, bend, rotate and undergo compressions efficiently [24,29]. This mechanical change in shape helps cells interact with the external world and perform the appropriate pattern of complex responses at the biochemical level [24,29].

The deformation of myofibril is recognized as especially paramount for the activation of several mechanotransductive signaling pathways that involve [28,29,30,31,32,33]:phosphatidylcholine-3-kinase (PI3K), which has a central role in controlling cellular growth [29,30];many proteins related to the mitogen-activated protein kinases (MAPKs), such as extracellular signal-regulated protein kinases 1 and 2 (ERK1/2) and c-Jun N-terminal kinase (JNK), which make metabolic and mechanical signals interact between them [29,30,32];the p38-MAPK pathway, which can change the shape of the cytoskeleton and can be regulated by mechanosensitive proteins, such as integrins (we shall focus on them below), located in the cell membrane [29,30,31,32];membrane phospholipids, which regulate how the cell membrane responds to external stimuli [30]. They can also affect the energetic metabolism of muscle cells through 5′ adenosine monophosphate-activated protein kinase (AMPK) [28,31] and glycogen-synthase [33].

At the biochemical level, Ca^2+^ ions play a leading role in the processes mentioned above [34]. Indeed, Ca^2+^ is involved in many intracellular activities, including mitochondrial development, plasticity and respiration [35], which constitute the present paper’s focus and will be discussed in the next section.

Unfortunately, despite the evidence on the role of mechanobiology in muscles, there is a lack of knowledge about the mechanosensitive properties of muscle cells, in particular, cell nuclei. The reason may lie in the fact that the more advanced investigative techniques used in the mechanobiological field have not been introduced in exercise physiology [29]. The usual methods used to apply mechanical stimuli to muscle cells in vivo or in vitro do not allow a thorough understanding of the observed phenomena, including the increase in strength after training or in flexibility after stretching exercises [30]. A deeper comprehension would be permitted by more sophisticated techniques such as tridimensional cytometry or scanning ion-conductance microscopy (SICM), which have the potential of assessing the mechanobiological behavior of muscle cells on nanometric size—that is, on mitochondrial size [29,30,36].

## 3. The Role of Mitochondria in Muscles and Physical Activity

### 3.1. The Mitochondria at the Center of the Organism

Mitochondria are the cellular organelles that produce adenosine triphosphate (ATP), the primary energy substrate for the human organism. Derived from bacteria engulfed by what became the first eukaryotic cell, mitochondria contain their DNA (mtDNA) and are involved in many metabolic processes [37,38,39]:amino acids, nucleotides, porphyrin, cholesterol, steroid hormones, glutathione and nitric oxide synthesis [38,39];antioxidant and ROS production and regulation [37,38];intra- and extracellular signaling through the release of Ca^2+^ and ROS [37,38,39];cell survival regulation, since mitochondria affect apoptosis [37,38].

Due to the functions just mentioned and the discovery that mitochondria are involved in pathologies affecting several organic systems (e.g., metabolic, cardiovascular, autoimmune, and neurological), researchers have recognized mitochondria among the main factors that allow the organism to face environmental stimuli through the stress response, whether local or systemic [39,40,41]. Indeed, mitochondria can regulate the organic adaptation by affecting both homeostasis—the maintenance of physiological parameters within a determined range beyond which irreversible damages and death would occur—and allostasis—the spontaneous and proactive change of physiological parameters to face environmental stressors [39,40,41].

The most recent research on stress pathophysiology has focused on the brain and the endocrine and immune systems [40], and it has investigated various biomarkers to comprehend the effects stress has on the body [39,41]. The conducted studies have highlighted that mitochondria can mediate how adverse psychosocial experiences impact on cellular and subcellular functions such as the stimulation and inhibition of immune and inflammatory response, oncogenic processes, gene regulation, and telomere maintenance [41]. In particular, mitochondria seem to have a double role in translating the effects of the stressful experience, an example being physical training. On the one hand, mitochondria could be the main target of the stress response; on the other hand, they could affect subsequent cellular or molecular alterations, thus acting as mediators of the stress response [39,40,41].

Furthermore, mitochondria play a central role in the stress response due to their ability to produce energy: in fact, the organism requires an increase in energy production to perform both the stress response and allostatic processes [40]. The more the energy produced, the better the brain can redistribute it to each organ or system that needs it to face the stressors efficiently. Indeed, this energy helps organs and cells change their structure and function based on the neuroendocrine molecules released during the stress response [39,40,41]. These signaling molecules can then induce epigenetic modifications or activate various cellular signaling pathways to direct structural and functional changes. In living organisms, energy may be essentially chemical, e.g., ATP obtained through oxidative phosphorylation, or thermic, such as the energy released through the expression of uncoupling proteins (UCPs) or due to the Brownian motion of molecules [40]. Mitochondria are paramount for energetic metabolism since both ATP production and UCPs thermogenic activity occur inside them [40].

Among environmental stressors, surely physical activity represents one of the stimuli that may strain the organism the most, both physically and psychologically—think about the high pressure that athletes, in particular elite ones, need to endure during training or the athletic performance [42]. At the muscular level, mitochondria may go through rapid and peculiar modifications—changing in volume, protein expression, and used substrate—based on muscle activity and cell environmental conditions, such as concentrations of nutrients, ROS, or lactic acid [43,44]. Indeed, skeletal muscle tissue requires a different amount of oxygen based on the sustained physical load, and mitochondria undergo the modifications mentioned above accordingly to provide muscles with the necessary oxygen [45].

It is noteworthy that “a mitochondrion is not an island:” inside cells, these organelles create dynamic networks throughout the cytoplasm to allow cells to adapt efficiently. In particular, mitochondria can connect to create more prominent cellular structures—mitochondrial fusion—or split in smaller organelles—mitochondrial fission. In addition, mitochondria may die—mitophagy—or be born anew—biogenesis—based on the stimuli received from the environment and, especially, from the cell nucleus [38,40,46].

### 3.2. Mitochondrial Biogenesis and Physical Activity

Mitochondrial biogenesis is among the biological processes that physical activity influences the most. Indeed, a prolonged physical exercise characterized by high oxygen consumption requires a great increase in the number of available mitochondria, which must also show a heightened oxidative enzymatic capacity for producing the required energy [46].

Whereas several factors regulate mitochondrial biogenesis during physical activity, others may induce mitophagy during immobilization [46]. These factors are molecules and intracellular signaling pathways involved in different cellular responses that can influence the cellular energetic metabolism [46]. Specifically:in the case of muscular contraction, biogenesis is promoted by an increase in ROS, intracellular Ca^2+^ flux, the kinases p38-MAPK and AMPK [37,46]. These are central to regulating the hunger–satiety balance and the consumption and creation of glucose/glycogen reserve. They also can activate or inhibit the intracellular messenger cyclic adenosine monophosphate (cAMP), which is induced by many hormones and neuropeptides, including catecholamines [46]. Biogenesis is also upregulated by an increased expression of sirtuins, which are proteins related to longevity, and CREB transcription factor, which is involved in neuronal plasticity and memory formation [28,46]. All these factors induce the synthesis of the peroxisome proliferation factor PGC-1α that, as a consequence, stimulates mitochondrial biogenesis [28,46]. It is worth noting that peroxisomes—the other organelles whose generation can be promoted by PGC-1α—interact strongly with mitochondria to carry out several vital cellular functions [47]. Indeed, they cooperate in the disposal of toxins or metabolic waste, in the regulation of ROS production, in the activation of immune response, and in the cholesterol metabolism, and in the β-oxidation of fatty acids [28,47];in case of immobilization, mitophagy is promoted by a reduction in the synthesis of PGC-α, sirtuins, AMPK and insulin-like growth factor 1 (IGF-1) [28,46]. As a result, the forkhead box O-class (FOXO) transcription factors become active and induce a cascade of several intracellular pathways that lead to protein degradation and mitophagy [46].

It is worth noting that during high-intensity training, FOXO factors interact with a high AMPK production to assure the processes of protein degradation and mitochondrial biogenesis could work together for allowing muscles the best possible remodeling and growth [46]. On the contrary, too much physical effort makes protein degradation prevail over biogenesis, thus weakening musculature [46].

Endurance training, which is mainly aerobic, represents the kind of physical activity that affects mitochondrial biogenesis the most by stimulating especially the activator protein 1 (AP-1) transcription factor, which is involved in oxidation, inflammation and apoptosis [45]. Endurance training also enables the expression of AMPK, PGC-1α and the peroxisome proliferator-activated receptors α and γ (PPAR-α and PPAR-γ), both of which regulate cellular oxidation [45,46,48]. On the other hand, resistance or anaerobic training leads mainly to muscular hypertrophy, that is, an increase in muscle volume due to a higher synthesis of muscle fibers, collagen fibers, and other proteins paramount for muscular metabolism. Muscular hypertrophy is regulated by different cellular pathways, in particular, by the anabolic PI3K–Akt–mTOR pathway [48]. Despite these different consequences, both aerobic and anaerobic exercises share several metabolic pathways and, as a result, even anaerobic training may increase the PGC-1α synthesis [49].

Regarding endurance training, the available literature suggests that the cellular factors mentioned above could also induce the transcription of proteins involved in the Krebs cycle and respiratory chain, thus influencing energy production [45]. Nonetheless, since these factors participate in several cellular responses, they could be induced either by metabolic stimuli—diet (i.e., antioxidants, fasting and carbohydrates, proteins or fat consumption), inflammation, cellular growth factors and stress hormones—or mechanical stimuli (Figure 1) [37,39,45,46].

Therefore, the present paper shall now focus on how the mechanical dimension may contribute to the mitochondria processes outlined in the last paragraphs.

### 3.3. Mitochondria as Mechanosensitive Organelles

Among the mechanical stimuli that could affect the cellular factors involved in mitochondrial biogenesis, we may include muscle contractions and stretching, the stiffness or elasticity of the extracellular matrix (ECM), and the mechanical forces created by other cells and organelles [37]. All these mechanical stimuli may be transmitted to the mitochondrial network through the cytoskeleton and its components, that is, actin filaments, microtubules and intermediate filaments [37,45]. Indeed, the cytoskeleton sustains, connects and shapes each cellular structure—without the cytoskeleton, cells and organelles would break apart [37,45].

Mitochondria are linked with the cytoskeleton in several ways. For instance, mitochondria contain tubulin, a protein included in the structure of cytoskeletal microtubules; they are attached to vimentin, another protein that composes intermediate filaments, without which they would disintegrate; lastly, they move on actin filaments [37].

The forces applied on the cytoskeletal filaments could thus induce a spatial remodeling that moves mitochondria closer to or away from each other [37,45]. As a result, mechanical forces facilitate mitochondria fusion or fission, whose alternation is paramount to regulating muscle metabolism finely and allowing muscles to consume the energy needed for facing the actual stressor [37,45]. Furthermore, a correct balance between fusion and fission favors biogenesis: on the one hand, fission facilitates the mitophagy of defective mitochondria [37,50]; on the other hand, fusion facilitates Ca^2+^ influx, which constitutes one of the most critical signals for the birth of new functional mitochondria [37] (Figure 2). The correct balance between fusion and fission also allows cells to distribute their mitochondria to their daughter cells during mitosis [37,45,46,50].

As reported in the scientific literature over the last years, changes in the cytoskeletal mechanical tension could affect mitochondria since these organelles and their ionic channels—the same channels that make Ca^2+^ enter the mitochondrial matrix—are mechanosensitive structures [51,52,53]. In fact, due to changes in mechanical tension, the channels located in the mitochondrial membrane can open or close and, as a result, alter the cellular metabolism [36,51,52,53].

In addition, cytoskeletal filaments are not merely attached to the mitochondrial membrane: they are linked directly with the ionic channels lying in the mitochondrial membrane, in particular, with the voltage-dependent anion channels (VDACs) [37]. VDACs allow the flux of ATP, adenosine diphosphate (ADP), pyruvate, malate, and other molecules paramount for the energetic metabolism [54]. VDACs also determine the mitochondrial membrane potential, a characteristic that can influence the mitochondrial function heavily [37]. Therefore, changes in the cytoskeletal tension can alter the mitochondrial membrane potential, directly impacting mitochondria life and death [37,54].

### 3.4. Mitochondrial Mechanotransduction as a Means for Biological Adaptation

The mechanical stimuli that occur in the body include transient, monotonous and fluctuation (e.g., muscle contractions, heartbeat and breathing) stress [37]. All these mechanical stimuli can affect mitochondrial structure and function: on the one hand, both transient and monotonous mechanical stretches can increase ROS production and facilitate mitochondrial fission. On the other hand, fluctuations in shear stress or cycle-by-cycle stretch can modify the cytoskeletal architecture, bioenergetic metabolism and expression of intracellular signaling pathways [37].

Analyzing ATP production by vascular smooth muscle cells (VSMCs), researchers found an almost linear correlation between mechanical stress, mitochondrial fractal dimension (FD) and ATP production [37]. FD is an index that measures mitochondria’s ability to occupy space: the higher the FD, the more complex the mitochondrial network and the more energy produced. In particular, it was found that [37]:under no mechanical stress, FD and ATP production are both low;under continuous monotonous stress (e.g., 4 h of static cellular stretch), both FD and ATP increase;under variable stress (e.g., 4 h of cyclic stretch of variable intensity), FD increases considerably, as proof of mitochondrial fusion and biogenesis, and ATP production may rise even to 10 times the resting value (i.e., when mechanical stress is absent).

As the deformation cells undergo during physiological function of tissues represents a paramount mechanical stimulus, even ECM stiffness represents a fundamental source of mechanical stress [37]. ECM is the extracellular environment in which cells live; it is made of ground substance, collagen, elastin fibers, and many proteins (e.g., fibronectin, laminin, and vimentin) the cytoskeleton attaches to through its filaments [37]. Furthermore, even the cell membrane attaches to ECM fibers through several components, including integrins, which are maybe the mechanosensitive proteins par excellence [12,37] (Figure 2).

Inside cells, many macromolecular protein complexes—the focal adhesions—link the ECM, cytoskeleton and cell membrane to each other through integrins [12]. As a result, these structures function as an integrated network in which chemical and mechanical information are shared and transmitted [12,14]. A deformation occurring inside the ECM may thus spread to the cytoskeleton and then to mitochondria, altering their structure and function [37].

Unfortunately, to the best of our knowledge, just a few studies have investigated the influence ECM might have on mitochondria. Nonetheless, there is evidence that, in VSMCs, ECM stiffness may affect mitochondrial structure and function, in particular, mitochondrial dimensions and energy production [37]. Moreover, in cardiomyocytes, ECM stiffness may affect basal metabolism and, by interacting with the spatial direction actin fibers follow, the cells’ ability to adapt to stressors, which depends on mitochondrial activity [55].

### 3.5. The Importance of Ca^2+^Signaling for Muscles Cells

Among all intracellular signaling pathways, Ca^2+^ signaling undoubtedly plays a central to the muscle tissue.

Whereas voltage-gated calcium channels (Ca_V_) are located in the membrane of many different cells, skeletal muscle cells express a particular channel that behaves differently compared to all other Ca_V_s: the channel Ca_V_1.1 [56]. This channel does not induce Ca^2+^influx in the cell when activated; instead, whether activated by an electric or mechanical stimulus, Ca_V_1.1 induces the release of the intracellular Ca^2+^ stored in the sarcoplasmic reticulum to begin muscle contraction [56].

Concerning the muscle sarcoplasmic reticulum, it is worth noting that it constitutes a system similar to the endoplasmic reticulum (ER) of the other eukaryotic cells [57,58]. ER has been found to have a peculiar link with mitochondria due to being distant just a few nanometers from them. Indeed, ER and mitochondria communicate through contact sites called mitochondria-associated ER membranes (MAMs) [57,58]. MAMs allow a direct passage for molecules to travel between ER and mitochondria: as a result, mitochondrial activity is finely regulated [57,58]. In addition, MAMs allow Ca^2+^ flux inside mitochondria whenever needed, thus facilitating muscle contraction, and regulating several cellular processes, such as mitochondrial fission, inflammasome formation (a particular protein complex released when cells sense potentially dangerous substances) and cellular autophagy [37,57,58]. The ER constitutes thus a paramount reservoir for Ca^2+^ ions that can be released upon electric, chemical or mechanical stimulation.

Returning to skeletal muscle fibers, Ca^2+^ allows the neural control of muscle fibers: upon sensing a nerve impulse, Ca_V_s increase Ca^2+^ influx to generate an action potential that propagates along the cell membrane. Consequently, skeletal muscle contraction begins [35]. Ca^2+^ intracellular concentration also regulates intracellular processes such as the actin–myosin coupling required for muscle contraction, protein synthesis and degradation, and the fiber type shifting between fast-twitch and slow-twitch phenotypes [35]. The fiber type shifting is mediated, in particular, by Ca^2+^-sensitive proteases and transcription factors. Even mitochondrial respiration, plasticity and adaptations are affected by changes in Ca^2+^ concentration [35] (Figure 3).

The regulation of Ca^2+^ influx in muscle cells affects the contraction–relaxation cycle of muscle fibers, with significant consequences on the functions of many tissues, including blood vessels and bronchi [59,60]. Ca^2+^ influx also affects muscle tissue development since it influences neuromuscular junction formation and their functionality through muscle nicotinic acetylcholine receptors (nAChRs) [61]. The role Ca^2+^ ions play for skeletal muscle tissue integrity has been deemed so important that several studies investigated the correlation between Ca^2+^ signaling, muscle growth or hypertrophy (even in the heart) and muscle stem cells proliferation in both health and disease [34,62,63].

Ca^2+^ signaling affects ROS production inside muscle cells since ROS are formed during normal metabolic activity and muscle contractions [64]. Ca^2+^ ions also seem to be involved in glycolysis regulation as they influence the catalysis rate of some enzymes central to glycolysis. By modulating the energetic metabolism, Ca^2+^ ions can thus coordinate the beginning of muscle contractions [35].

Ca^2+^ influx in mitochondria increases the ability of these organelles to produce ATP by affecting mitochondrial crests and, thus, the electron transport chain. As a result, during muscle contraction, energetic homeostasis may be maintained more efficiently [35].

Lastly, Ca^2+^ ions participate in skeletal muscle hypertrophy, as this phenomenon represents an adaptive response subsequent to mechanical load. Although the mechanisms of action behind this effect have not yet been completely discovered, the main actors seem to be the molecule calcineurin and the ratio between extracellular ATP and intracellular Ca^2+^ [35,65].

Calcineurin is an intracellular Ca^2+^-dependent molecule that can affect the differentiation of skeletal muscles satellite cells (i.e., muscle stem cells) through epigenetic mechanisms. The satellite cell differentiation is vital for regenerating muscle fibers after lesions and guaranteeing hypertrophy in the long-term [35]. Concerning the ratio between extracellular ATP and intracellular Ca^2+^, an increase in plasmatic ATP (such as during physical activity) elicits a Ca^2+^ influx that induces the activation of the mTOR pathway and some MAPKs through complex signaling cascades central to cellular growth [65,66,67].

### 3.6. Ca^2+^Signaling as a Mechanosensitive and Mechano-Dependent Pathway

Even though Ca^2+^ channels have been usually viewed as dependent on chemical and electric stimuli, mechanobiology has shown that Ca^2+^ signaling is a mechanosensitive and mechano-dependent pathway [34,63]. Indeed, mechano-dependent Ca^2+^ channels are abundantly expressed in muscles during their whole development, hence fulfilling a crucial role for every muscle cell, whether cardiac, skeletal or smooth [34,63].

Ca^2+^ performs a pivotal role in transducing mechanical stimuli into intracellular signals as mechanical forces can change the cytosolic Ca^2+^ concentration and, as a result, induce the effects described in the previous paragraphs [63]. Indeed, mechanical tensions can affect the state of the membrane Ca^2+^-dependent channels by acting on the physical connections between the ECM, cytoskeleton and cell membrane (e.g., by stretching cytoskeletal filaments). Changes in intracellular Ca^2+^ concentration allow cells to acquire information about the external environment and to respond promptly by modifying their metabolism, gene expression and protein synthesis [34].

When undergoing mechanical stress, cells may change their cytoskeleton structure by remodeling actin filaments, whose functionality depends on Ca^2+^ availability. The cytoskeleton may facilitate Ca^2+^ influx through the opening of mechanosensitive stretch-activated ion channels (SACs): this event may then activate the membrane Ca^2+^ channels, with all due consequences. Therefore, the cytoskeleton could be viewed as a controller of cellular Ca^2+^ entry due to its mechanosensitivity and influence on SACs [34].

Lastly, among the several types of Ca^2+^ channels inside muscle cells, there are also the transient receptor potential (TRP) channels. TRPs have been vastly researched over the years due to their prominent role in mediating high threshold stimuli—TRP channels are central to nociception. In fact, they are also involved in mechanotransduction in many different tissues and cells: TRPs are sensitive to various mechanical forces, such as fluid shear stress and cell membrane stretching [34]. In particular, Ca^2+^ entry through TRP channels seems to be the main stimulus able to regulate the signaling pathways involved in muscle tissue regeneration and reparation [35].

## 4. Could Manual Therapies Be Mechanobiological and Mitochondrial Interventions?

The previous section has reviewed how muscle metabolism, mitochondrial functionality and Ca^2+^ influx can be affected by mechanical stimuli, in particular, physical activity. Since manual therapists apply mechanical forces to different body tissues to improve physiologic function [68], it is conceivable that athletic performance could be improved not only through personalized training but also by applying manual therapies. Indeed, manual therapies could act on muscle and connective tension, stiffness and elasticity: they might even affect ECM structure and function [68].

Hypothetically, if manual therapies could maintain ECM in an optimally functional state, both muscle cell membrane and cytoskeleton would be affected positively. It is our hypothesis that a functional (i.e., not too stiff) ECM could allow a more efficient opening and closure of ion channels, which could lead to a better Ca^2+^ flux regulation with positive consequences for cytoskeletal, mitochondrial, sarcoplasmic and muscle functionality and, therefore, for athletic performance.

Given these hypotheses, what does the available clinical research show? Can manual therapies induce changes that affect even mitochondrial biogenesis? Or is there the need to conduct further studies to test the mentioned hypotheses?

From a general perspective, various papers have shown that manual therapies or at least some forms of manual stimuli may influence cell functioning. In fact, the effects seem mechano-dependent (at least in the studies that performed a thorough analysis) since they do not occur when membrane integrins are inhibited by monoclonal antibodies [69].

Therefore, in the next paragraphs, we report the main effects on cell behavior discovered over the years: they were obtained by evaluating real manual therapies or using models that mimicked the application on organic tissues of mechanical stimuli similar to the manual ones. We tried to exclude from the following list the papers that used stimuli impossible to replicate in the clinical setting (e.g., stimuli applied for hours or even days).

Fibroblasts seem to enlarge and lengthen. They tend to develop more prominent nuclei and rough ER, maybe for sustaining an increased synthesis of collagen and proteoglycan [70].Keratinocytes seem to proliferate, and fibroblasts seem to align along the direction of the applied mechanical stimulus (pressure or traction). Cell metabolism seems to be affected as well: procollagen synthesis and collagen deposition rise whereas ECM degradation decreases [69].The links between fibroblasts and adjacent collagen fibers change: this event could allow the creation of new connections between the cytoskeleton filaments and the different ECM fibers [71].Fifteen minutes of massage on painful areas may change collagen disposition and liquid content in the derma and superficial fascia [72].Changes in cytokines and signaling molecules may occur. In the skin, cyclic stimuli seem to increase the secretion of interleukin-1β (IL-1β), prostaglandin-E2, angiotensin II and platelet-derived growth factor (PDGF) [69]. In muscle cells and fibroblasts, equibiaxial mechanical stimuli that last for less than five minutes appear to decrease inflammatory cytokines, including IL-1β, IL-3, IL-6, granulocyte colony-stimulating factor (GCSF) and macrophage-derived chemokine (MDC). On the contrary, uniaxial mechanical stimuli that last for several minutes appear to increase the production of the same cytokines mentioned above [73,74,75].In a trial that modeled an osteopathic technique (i.e., myofascial release), a mechanical stimulus applied for about one–two minutes on fibroblasts induced an increase in myoblast differentiation and regenerative capacity. A rise in nAChRs, which are paramount for efficient neuromuscular control, was also observed [75,76,77].From a more specific point of view, some papers assessed whether manual therapy could affect cell and mitochondrial behavior in muscles.In the case of quadriceps injury due to physical activity, compared to no treatment, massage seems to increase the expressions of mechanotransductive pathways (e.g., ERK1/2), focal adhesion kinases (FAKs) and PGC-1α, the main mediator of mitochondrial biogenesis. In addition, massage may decrease inflammatory cytokines (e.g., IL-6 and tumoral necrosis factor-α) and reduce the expression of nuclear factor κB, one of the main transcription factors involved in inflammation [7].A cyclic stimulus lasting for ten minutes applied on injured skeletal muscle cells (i.e., a model of muscular lesions) increased the expression of superoxide dismutase (SOD), which is one of the main body antioxidant enzymes, and reduced both malondialdehyde and creatine kinase (CK), which are typical markers of muscle oxidation and injury. The applied stimulus mimicked a myofascial technique used in Chinese Traditional Medicine. The just mentioned effects did not occur whenapplying a static and continuous stimulus; they were even suppressed by injecting a Ca^2+^ signaling antagonist in the cell culture. This last result could give support to the hypothesis that manual therapies might act through the activation of the Ca^2+^ signaling pathways described in the previous section [78].In rats suffering from gastrocnemius atrophy, a model mimicking a manual therapy protocol (i.e., 30 min of cyclic mechanical compressions a day, every other day, for four days) increased the transverse section and size of myofibrils, the rate of protein synthesis in myofibrils (but not in mitochondria), the activation of mechanosensitive FAKs and the proliferation of satellite cells. These effects could be viewed as strongly facilitating muscle regeneration [79].In rats suffering from gastrocnemius atrophy due to denervation, a model mimicking manual therapy slowed down the rate of muscle atrophy by increasing the expression of Akt, which is usually involved in cell survival and muscle anabolism, and reducing proteins usually related to muscle atrophy [80].

The just reported effects seem the main results that may be found by searching the literature about manual therapies, muscles and cell behavior.

Unfortunately, the papers analyzing this subject often appear to be of low quality or have low clinical usefulness (e.g., animal or in vitro studies, models distant from everyday practice). Moreover, there are also negative results: for instance, a study investigating 30 min of manual lymphatic drainage applied right after physical activity did not find any effects on muscle pain, CK and neutrophils levels compared to a rest condition [81]. Therefore, the relationship between manual therapies, muscles, cell behavior and athletic performance is a field that needs further and more sound research to be able to give clear and useful indications for clinical practice.

Indeed, based on the previous sections, the potential ability of manual therapies to affect muscle cell behavior through mechanobiological pathways could become particularly important for optimal management of athletes’ physical conditions. Furthermore, since inflammation may alter ECM structure and facilitate its stiffening, thus favoring fibrosis [82], manual therapies could play a crucial role in maintaining muscular health through their anti-inflammatory effects.

## 5. Conclusions

In the last decades, research in mechanobiology has significantly risen and, together with the knowledge coming from molecular biology, has allowed researchers to understand how mechanical stimuli could influence cell behavior. In fact, mechanical stimuli can even affect gene expression, thus influencing protein synthesis, cell metabolism and the functionality of mitochondria, the human organism’s powerhouses.

Knowing the cellular influence of mechanical stimuli allows us to widen our view on physical activity; in particular, athletic performance. Indeed, mechanobiology helps us understand better how muscles adapt to the loads they are exposed to during either aerobic or anaerobic training.

In addition, deepening the knowledge about mechanobiology would allow us to look differently at manual therapies—the primary intervention based on mechanical stimuli after physical exercise. Although there is a paucity of research on their microscopic effects, we have evidence that manual therapies could affect cell behavior and, potentially, mitochondrial functionality.

Therefore, we expect this review to inspire both researchers and clinicians to conduct further studies to test the extent to which manual therapies could affect tissue, cell and mitochondrial metabolism.

In particular, we expect laboratory researchers and biologists to cooperate with manual therapists to understand clearly the magnitude of the forces applied during manual therapies. As a result, researchers could create more realistic models that could be used in preclinical studies to understand better the effects of different manual therapies on different body tissues. The findings of these studies could help clinicians develop new preventive and therapeutic approaches, especially in the athletic field; they could also shed light on the clinical results, both local and systemic, that manual therapies manage to obtain [2].

Beyond the assessment of cytokines release, which is one of the main themes the research on manual therapies has focused on in the last years, we expect future in vitro studies to assess changes in the expression of mechanosensitive elements, including integrins, focal adhesions, gap junctions, desmosomes and Ca^2+^-dependent channels [83,84]. These studies could also assess changes in the expression of genes related to mitochondrial activity. Since manual therapies have shown to influence circulation [85,86], clinical studies could focus on seeking biomarkers related to mitochondrial activity even in blood and plasma samples (see [39,87] for possible biomarkers). Although metabolite analysis in blood is usually used to assess mitochondrial disorders, similar studies could assess whether manual therapies may change mitochondrial metabolism, especially in case of muscle injury.

Clinical studies should compare manual therapies with appropriate sham therapies to evaluate both the actual effects of manual stimuli and the effects of specific techniques. Such comparisons could also help understand whether light touch may be already able to elicit mechanotransductive responses.

In the athletic field, the results of the studies mentioned above would allow clinicians to choose the most appropriate manipulative treatment to help athletes recover from injuries. These studies could also help identify the most appropriate manual approaches able to prevent injuries. Indeed, athletes undergo several and different mechanical loads during the training season: knowing the mechano–chemical effects of manual techniques could allow therapists to accompany and support athletes in the best possible way during every stage of their training.

In conclusion, should the hypothesized studies be conducted and the effects of manual therapies on mitochondrial function be confirmed, healthcare and athletic performance management could be greatly improved.

## Figures and Tables

**Figure 1 healthcare-09-00181-f001:**
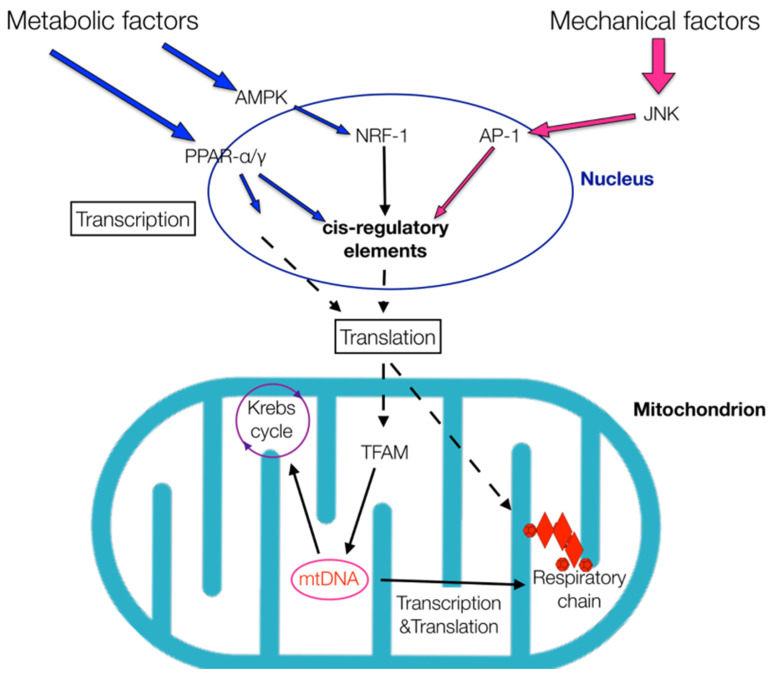
Metabolic and mechanical factors can affect energy production. Several signaling pathways—adenosine monophosphate-activated protein kinase (AMPK) and Jun N-terminal kinase (JNK)—and transcription factors—peroxisome proliferator-activated receptor (PPAR) α/γ, nuclear respiratory factor 1 (NRF-1) and activator protein 1 (AP-1)—are involved in this process. The transcription factors bind to various cis-regulatory elements that elicit the transcriptions of other neighboring genes. As a result, cell ribosomes synthesize proteins that influence mitochondrial metabolism by inducing mitochondrial transcription factors, such as the mitochondrial transcription factor A (TFAM). TFAM affects the expression of mitochondrial DNA (mtDNA) genes that induce enzymes necessary for both the Krebs cycle and respiratory chain, thus influencing energy production.

**Figure 2 healthcare-09-00181-f002:**
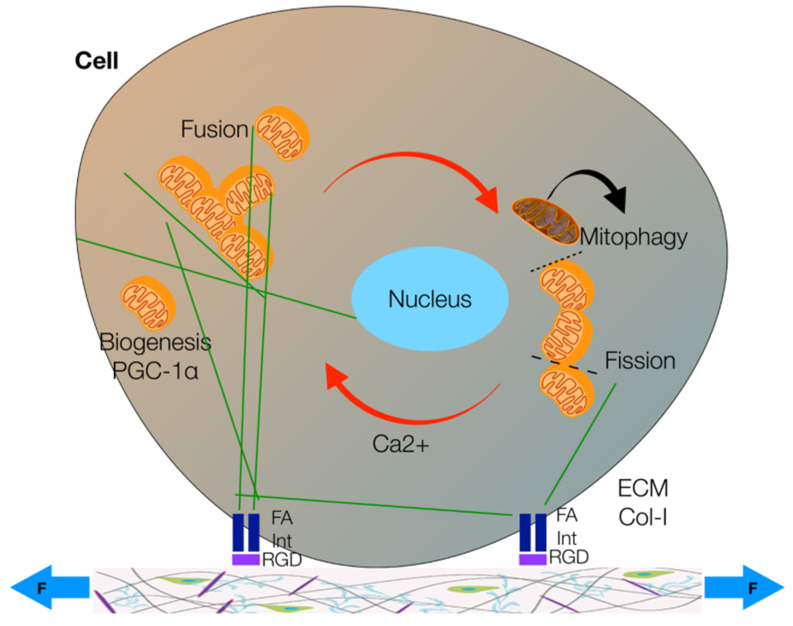
The processes of mitochondrial biogenesis (promoted by the peroxisome proliferation factor PGC-1α), mitophagy, fusion and fission. Mitochondria (orange) can move on the cytoskeletal filaments (green) to fuse in more complex structures or split in smaller organelles. The cytoskeleton and, thus, mitochondria’s motion and functionality are affected by extracellular forces (F) that, through the link between membrane integrin receptors (Int) and Arg-Gly-Asp (RGD) binding sites on extracellular matrix (ECM) collagen fibers (Col-I), can stimulate intracellular focal adhesion (FA). Ca^2+^ signaling is also paramount to mitochondrial fusion. Defective mitochondria (gray) undergo mitophagy (modified from [37]).

**Figure 3 healthcare-09-00181-f003:**
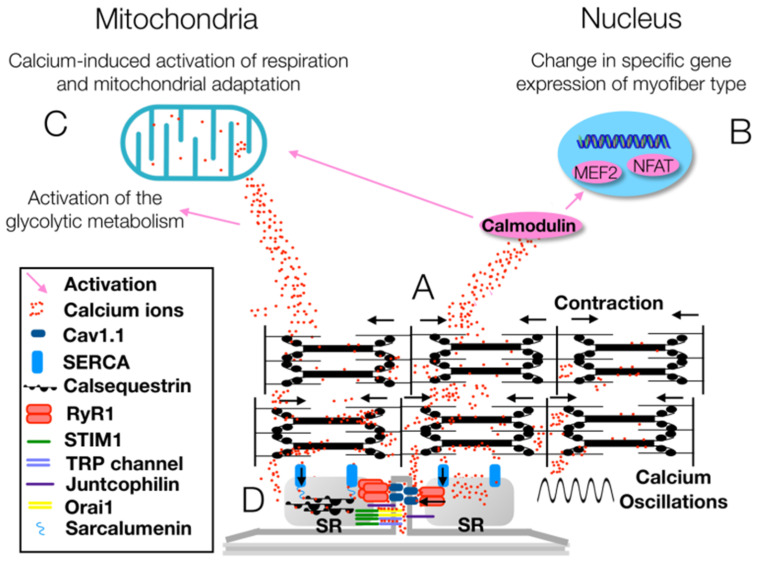
(**A**) The voltage-gated Ca^2+^ channel Ca_V_1.1 allows the release of Ca^2+^ from the sarcoplasmic reticulum (SR), whereas sarcoplasmic/endoplasmic reticulum Ca^2+^-ATPase (SERCA) pumps modulate the Ca^2+^ reuptake in the SR. As a result, muscle contraction and relaxation occur and, during high neuromuscular activity, the oscillation in Ca^2+^ levels induce an increase in Ca^2+^ concentration in myofibrils. (**B**) When Ca^2+^ levels increase in the sarcoplasm, a signaling cascade mediated by calmodulin upregulates the expression of the myocyte enhancer factor 2 (MEF2) and the nuclear factor of activated T-cells (NFAT), which favor the shift of skeletal muscle fibers towards the slow-twitch oxidative phenotype. (**C**) The signaling cascade mediated by calmodulin and the Ca^2+^ increase modulates the mitochondrial respiration and the production of glycolytic enzymes. (**D**) The stromal interaction module 1 (STIM1) protein senses decreasing Ca^2+^ levels in the SR and then gates both the Ca^2+^ release-activated Ca^2+^ channel protein 1 (Orai1) channel and the transient receptor potential (TRP) channel to increase Ca^2+^ concentration. Ca^2+^ concentration is regulated finely through the interaction of several proteins—junctophilin, sarcalumenin and calsequestrin—and channels—Ca_V_1.1, ryanodine receptor 1 (RyR1) and SERCA (modified from [35]).

## Data Availability

Not applicable.

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
