# Peer review of "Sport Performance and Manual Therapies: A Review on the Effects on Mitochondrial, Sarcoplasmatic and Ca2+ Flux Response"

_healthcare, 2021, doi:10.3390/healthcare9020181_

Round 1

Reviewer 1 Report

thank you for permitting me to review this manuscript 

Abstract 

the abstract should totally be rewritten  although an abstract should summerize the work but it should also encourage to read the whole manuscript , right now it does not encourage to read the manuscript because the conclusion is negative in addition the aims of the review appearss only in the middle of the abstract , it should appear in the begining.

Introduction 

  A one sentence definition of manual therapy should appear in the first paragraph of introduction.

Please annotate appropriately in the introduction mecanobiology appears as 1) but right after it appears as 2 which might be confusing it can be a sub chapter of introduction or methods 

Wolf law should be defined quickly  for non experts readers if its too long in an appendix

page 3 line 3 to 17 please relate the exact references to point annotations and subsequently in other paragraphs in the whole manuscript 

figure 1 

should have bigger character it is not readable 

page 10 chapter 4

could manual therapies .....

the first 3 paragraphs , please add references

conclusion 

please be more elicit in designing further directions of research for clinical and laboratory researchers 

there is no negative conclusion as cited in the abstract please be coherent with the conclusion of the abstract 

Author Response

Cover letter Reviewer 1

We thank the reviewer for his precious comments.

We tried to respond in the best way we could.

We also review the English of our paper as advised by both reviewers.

the abstract should totally be rewritten  although an abstract should summerize the work but it should also encourage to read the whole manuscript , right now it does not encourage to read the manuscript because the conclusion is negative in addition the aims of the review appearss only in the middle of the abstract , it should appear in the begining.

We revised the abstract as follows:

The present narrative review aims to highlight the possible effects manual therapies could have on cells and mitochondria, as these effects could improve athletic performance management. To his aim, this review summarizes the relationship between mechanical stimulation, with a special focus on physical activity, and cell response based on the most recent mechanobiology findings.

Mechanobiology analyzes how cells respond to mechanical stressors coming from the environment. Indeed, endogenous (e.g., blood pressure, heartbeat and gastrointestinal motility) and exogenous (e.g., physical activity and manual therapies) stimuli can induce biochemical and epigenetic modifications that alter protein synthesis with heavy consequences on cell behavior. Mechanical stress can also influence mitochondrial behavior (i.e., biogenesis, autophagy, fusion, fission and energy production), sarcoplasmic response and calcium ion (Ca2+) flux.

Since manual therapies have been shown to affect the extracellular matrix, which represents a primary source of mechanical stress that may alter both the cytoskeleton and mitochondrial metabolism, it is conceivable manual therapies could also affect cellular and mitochondrial behavior.

Lastly, by suggesting possible directions for future laboratory and clinical studies, the authors expect this review to inspire further research on how manual therapies could affect bioenergetic metabolism and, thus, athletic performance.

A one sentence definition of manual therapy should appear in the first paragraph of introduction.

We added the following paragraph at the beginning of the Introduction (Page 1):

Manual therapies are a form of treatment delivered by healthcare professionals through their hands to evaluate and treat functional disorders or pathologies [1]. Specific manual techniques are used to engage every body structure, including articulations, myofascial structures, nerve pathways, and circulation. Manual therapeutic approaches such as osteopathic manipulative treatment have shown to be able to improve clinical conditions in both adults and children, sometimes even in a way comparable to standard care [1,2].

New references:

  1. Smith, M.S.; Olivas, J.; Smith, K. Manipulative Therapies: What Works. Am. Fam. Physician 2019, 99, 248–252.
  2. Field, T. Massage Therapy Research Review. Complement. Ther. Clin. Pract. 2016, 24, 19–31, doi:10.1016/j.ctcp.2016.04.005.

Please annotate appropriately in the introduction mecanobiology appears as 1) but right after it appears as 2 which might be confusing it can be a sub chapter of introduction or methods 

We struggled to understand what the reviewer meant with this comment. We guessed that maybe the reviewer referred to the last paragraph of the Introduction – where we say that the first part of our paper will deal with mechanobiology – and to the section right after the Introduction – whose chapter number is “2”. Since we followed the manuscript template made by Healthcare, which numbers the Introduction as “1”, we opted for a bullet list to describe our paper organization (Page 2, line 20).

Therefore, the present review will discuss the importance of the mechanobiological dimension for muscles, the bioenergetic role of mitochondria, and the mechanosensitive properties of these organelles. Lastly, the main cellular effects showed by manual therapies in preclinical and clinical studies will be described.

Wolf law should be defined quickly  for non experts readers if its too long in an appendix

We added the following paragraph (Page 2, line 47):

A clear example of the importance of mechanobiology is Wolff’s Law, despite its limitations that have been overcome in the years. Developed in 1892 by the German physician Julius Wolff, the law states that bones adapt to mechanical loads by streng-thening their internal and external architecture as mechanical loads increase. On the contrary, as mechanical loads decrease, bone resorption shall occur thus leading to os-teopenia or osteoporosis [20]. According to Wolff, bone remodeling depended essentially on how mechanical loads were applied, that is, on their duration, intensity and velocity. Today, we know that bone remodeling occurs since mechanical stimuli are translated into cellular biochemical cascades through the involvement of mechanotransducing processes [21,22].

New references:

  1. Mullender, M.G.; Huiskes, R. Proposal for the Regulatory Mechanism of Wolff’s Law. J. Orthop. Res. 1995, 13, 503–512, doi:10.1002/jor.1100130405.
  2. Pearson, O.M.; Lieberman, D.E. The Aging of Wolff’s ?Law?: Ontogeny and Responses to Mechanical Loading in Cortical Bone. Am. J. Phys. Anthropol. 2004, 125, 63–99, doi:10.1002/ajpa.20155.
  3. Spyropoulou, A.; Karamesinis, K.; Basdra, E.K. Mechanotransduction Pathways in Bone Pathobiology. Biochim. Biophys. Acta BBA - Mol. Basis Dis. 2015, 1852, 1700–1708, doi:10.1016/j.bbadis.2015.05.010.

page 3 line 3 to 17 please relate the exact references to point annotations and subsequently in other paragraphs in the whole manuscript 

We point the references as follows (Page 3, line 18):

The deformation of myofibril is recognized as especially paramount for the activation of several mechanotransductive signaling pathways that involve [28–33]:

  • phosphatidylcholine-3-kinase (PI3K), which has a central role in controlling cellular growth [29,30];
  • many proteins related to the mitogen-activated protein kinases (MAPKs), such as extracellular signal-regulated protein kinases 1 and 2 (ERK1/2) and c-Jun N-terminal kinase (JNK), which make metabolic and mechanical signals interact between them [29,30,32];
  • the p38-MAPK pathway, which can change the shape of the cytoskeleton and can be regulated by mechanosensitive proteins, such as integrins (we shall focus on them below), located into the cell membrane [29–32];
  • membrane phospholipids, which regulate how the cell membrane responds to ex-ternal stimuli [30]. They can also affect the energetic metabolism of muscle cells through 5’ adenosine monophosphate-activated protein kinase (AMPK) [28,31] and glycogen-synthase [33].

As advised, we did the same for the whole manuscript (e.g., page 4, line 1).

figure 1: should have bigger character it is not readable 

We increased the size of almost every label: we hope they are now more easily readable.

page 10 chapter 4 could manual therapies .....: the first 3 paragraphs , please add references

Since the first three paragraphs of this chapter briefly resume the topics described in the precedent sections and contain just our hypotheses, there are not actual references to add. Nonetheless, we revised the paragraphs to make readers understand better the hypothetical character of our discussion, and we also added some potentially useful reference (page 10, line 42).

The previous section has reviewed how muscle metabolism, mitochondrial functionality and Ca2+ influx can be affected by mechanical stimuli, in particular, physical activity. Since manual therapists apply mechanical forces to different body tissues to im-prove physiologic function [68], it is conceivable that athletic performance could be im-proved not only through personalized training but also by applying manual therapies. Indeed, manual therapies could act on muscle and connective tension, stiffness and elas-ticity: they might even affect ECM structure and function [68].

Hypothetically, if manual therapies could maintain ECM in an optimally functional state, both muscle cell membrane and cytoskeleton would be affected positively. It is our hypothesis that a functional (i.e., not too stiff) ECM could allow a more efficient opening and closure of ion channels, which could lead to a better Ca2+ flux regulation with positive consequences for cytoskeletal, mitochondrial, sarcoplasmic and muscle functionality and, therefore, for athletic performance.

Given these hypotheses, what does the available clinical research show? Can manual therapies induce changes that affect even mitochondrial biogenesis? Or is there the need to conduct further studies to test the mentioned hypotheses?

New reference:

  1. Tozzi, P. A Unifying Neuro-Fasciagenic Model of Somatic Dysfunction – Underlying Mechanisms and Treatment – Part II. J. Bodyw. Mov. Ther. 2015, 19, 526–543, doi:10.1016/j.jbmt.2015.03.002.

please be more elicit in designing further directions of research for clinical and laboratory researchers 

We revised the conclusion adding several suggestions for further studies (Page 12, line 49):

In particular, we expect laboratory researchers and biologists to cooperate with manual therapists to understand clearly the magnitude of the forces applied during manual therapies. As a result, researchers could create more realistic models that could be used in preclinical studies to understand better the effects of different manual therapies on different body tissues. The findings of these studies could help clinicians develop new preventive and therapeutic approaches, especially in the athletic field; they could also shed light on the clinical results, both local and systemic, that manual therapies manage to obtain [2].

Beyond the assessment of cytokines release, which is one of the main themes the research on manual therapies has focused on in the last years, we expect future in vitro studies to assess changes in the expression of mechanosensitive elements, including integrins, focal adhesions, gap junctions, desmosomes and Ca2+-dependent channels [83,84]. These studies could also assess changes in the expression of genes related to mitochondrial activity. Since manual therapies have shown to influence circulation [85,86], clinical studies could focus on seeking biomarkers related to mitochondrial activity even in blood and plasma samples (see [39,87] for possible biomarkers). Although metabolite analysis in blood is usually used to assess mitochondrial disorders, similar studies could assess whether manual therapies may change mitochondrial metabolism, especially in case of muscle injury.

Clinical studies should compare manual therapies with appropriate sham therapies to evaluate both the actual effects of manual stimuli and the effects of specific techniques. Such comparisons could also help understand whether light touch may be already able to elicit mechanotransductive responses.

In the athletic field, the results of the studies mentioned above would allow clinicians to choose the most appropriate manipulative treatment to help athletes recover from injuries. These studies could also help identify the most appropriate manual approaches able to prevent injuries. Indeed, athletes undergo several and different mechanical loads during the training season: knowing the mechano-chemical effects of manual techniques could allow therapists to accompany and support athletes in the best possible way during every stage of their training.

In conclusion, should the hypothesized studies be conducted and the effects of manual therapies on mitochondrial function be confirmed, healthcare and athletic performance management could be greatly improved.

there is no negative conclusion as cited in the abstract please be coherent with the conclusion of the abstract 

Having revised the abstract as suggested, the conclusions should now be coherent with the abstract.

Reviewer 2 Report

Dear authors,

I did enjoy reading your interesting review on the roles of mitochondria in the response to mechanical triggers. I did like the bullet point summaries - I think they work very well making the information easily accessible.

Here are my few comments:

Title: I wonder whether the inclusion of "Exercise" may be better and whether manual therapy may be more suitable than manual medicine to describe the impact on sporting practice (eg Exercise Performance and manual therapy: a review on the effects on mitochondrial, sarcoplasmatic and Ca2+ flux responses")

page 2: please introduce Wolff´s law to the reader

page 5 top: on the topic of the peroxisome-mitochondria interactions, it may be worth adding that both organelles cooperate in the metabolism of fatty acids through the beta-oxidation pathways (peroxisomes: long and branched FA, mitos: medium-short FA, interaction via eg MDVs mitochondria derived vesicles see for example Cell Biosci. 2019 Mar 19;9:27. doi: 10.1186/s13578-019-0289-8.)

Some of the labels may be too small in Figs 1, 2 and 3 - please check whether all labels can be read in their final size (eg panel D in Fig 3)

Fig 2. it may be helpful to include "Ca2+" under the lower, left going arrow to indicate the positive impact of calcium on mitochondrial fusion as stated in the main text

Author Response

We thank the reviewer for his precious comments.

We tried to respond in the best way we could.

We also review the English of our paper as advised by both reviewers.

Title: I wonder whether the inclusion of "Exercise" may be better and whether manual therapy may be more suitable than manual medicine to describe the impact on sporting practice (eg Exercise Performance and manual therapy: a review on the effects on mitochondrial, sarcoplasmatic and Ca2+ flux responses")

We thank the reviewer for the suggestion. After having discussed it, we chose to revise the title as “Sports Performance and manual therapies: a review on the effects on mitochondrial, sarcoplasmatic and Ca2+ flux responses"

page 2: please introduce Wolff´s law to the reader

We added the following paragraph (Page 2, line 47):

A clear example of the importance of mechanobiology is Wolff’s Law, despite its limitations that have been overcome in the years. Developed in 1892 by the German physician Julius Wolff, the law states that bones adapt to mechanical loads by strengthening their internal and external architecture as mechanical loads increase. On the contrary, as mechanical loads decrease, bone resorption shall occur thus leading to osteopenia or osteoporosis [20]. According to Wolff, bone remodeling depended essentially on how mechanical loads were applied, that is, on their duration, intensity and velocity. Today, we know that bone remodeling occurs since mechanical stimuli are translated into cellular biochemical cascades through the involvement of mechanotransducing processes [21,22].

New references:

  1. Mullender, M.G.; Huiskes, R. Proposal for the Regulatory Mechanism of Wolff’s Law. J. Orthop. Res. 1995, 13, 503–512, doi:10.1002/jor.1100130405.
  2. Pearson, O.M.; Lieberman, D.E. The Aging of Wolff’s ?Law?: Ontogeny and Responses to Mechanical Loading in Cortical Bone. Am. J. Phys. Anthropol. 2004, 125, 63–99, doi:10.1002/ajpa.20155.
  3. Spyropoulou, A.; Karamesinis, K.; Basdra, E.K. Mechanotransduction Pathways in Bone Pathobiology. Biochim. Biophys. Acta BBA - Mol. Basis Dis. 2015, 1852, 1700–1708, doi:10.1016/j.bbadis.2015.05.010.

page 5 top: on the topic of the peroxisome-mitochondria interactions, it may be worth adding that both organelles cooperate in the metabolism of fatty acids through the beta-oxidation pathways (peroxisomes: long and branched FA, mitos: medium-short FA, interaction via eg MDVs mitochondria derived vesicles see for example Cell Biosci. 2019 Mar 19;9:27. doi: 10.1186/s13578-019-0289-8.)

We thank the reviewer for the suggestion. We added the suggested reference and revised the paragraph as follows (Page 5, line 23):

It is worth noting that peroxisomes – the other organelles whose generation can be promoted by PGC-1α – interact strongly with mitochondria to carry out several vital cellular functions [47]. Indeed, they cooperate in the disposal of toxins or metabolic waste, in the regulation of ROS production, in the activation of immune response, and in the cholesterol metabolism, and in the β-oxidation of fatty acids [28,47];

Some of the labels may be too small in Figs 1, 2 and 3 - please check whether all labels can be read in their final size (eg panel D in Fig 3)

We increased the size of almost every label: now, they should be more easily readable.

Fig 2. it may be helpful to include "Ca2+" under the lower, left going arrow to indicate the positive impact of calcium on mitochondrial fusion as stated in the main text

We included the label “Ca2+” as the reviewer suggested: we hope to have put it in the right place.

Round 2

Reviewer 1 Report

The authors have  responded adequately point by point to the comments of the review and have improved consequently the manuscript.